# A Nomogram for Predicting Non-Response to Surgery One Year after Elective Total Hip Replacement

**DOI:** 10.3390/jcm11061649

**Published:** 2022-03-16

**Authors:** Michelle M. Dowsey, Tim Spelman, Peter F. M. Choong

**Affiliations:** 1Department of Surgery, The University of Melbourne, St. Vincent’s Hospital Melbourne, Fitzroy, VIC 3065, Australia; tim@burnet.edu.aup (T.S.); choong@unimelb.edu.au (P.F.M.C.); 2Department of Orthopaedics, St. Vincent’s Hospital Melbourne, Fitzroy, VIC 3065, Australia

**Keywords:** hip replacement, outcome, risk prediction, nomogram

## Abstract

Background: Total hip replacement (THR) is a common and cost-effective procedure for end-stage osteoarthritis, but inappropriate utilization may be devaluing its true impact. The purpose of this study was to develop and test the internal validity of a prognostic algorithm for predicting the probability of non-response to THR surgery at 1 year. Methods: Analysis of outcome data extracted from an institutional registry of individuals (N = 2177) following elective THR performed between January 2012 and December 2019. OMERACT-OARSI responder criteria were applied to Western Ontario and McMaster Universities Arthritis Index (WOMAC) pain and function scores at pre- and 1 year post-THR, to determine non-response to surgery. Independent prognostic correlates of post-operative non-response observed in adjusted modelling were then used to develop a nomogram. Results: A total of 194 (8.9%) cases were deemed non-responders to THR. The degree of contribution (OR, 95% CI) of each explanatory factor to non-response on the nomogram was, morbid obesity (1.88, 1.16, 3.05), Kellgren–Lawrence grade <4 (1.89, 1.39, 2.56), WOMAC Global rating per 10 units (0.86, 0.79, 0.94) and the following co-morbidities: cerebrovascular disease (2.39, 1.33, 4.30), chronic pulmonary disease (1.64; 1.00, 2.71), connective tissue disease (1.99, 1.17, 3.39), diabetes (1.86, 1.26, 2.75) and liver disease (2.28, 0.99, 5.27). The concordance index for the nomogram was 0.70. Conclusion: We have developed a prognostic nomogram to calculate the probability of non-response to THR surgery. In doing so, we determined that both the probability of and predictive prognostic factors for non-response to THR differed from a previously developed nomogram for total knee replacement (TKR), confirming the benefit of designing decision support tools that are both condition and surgery site specific. Future external validation of the nomogram is required to confirm its generalisability.

## 1. Introduction

Total hip joint replacement (THR) continues to be one of the most common and cost-effective surgeries performed in the history of orthopaedics and medicine [1] and its popularity for treating end-stage osteoarthritis is reflected in over 50,000 procedures that are performed each year in Australia [2]. Estimates for Australia now predict a rise of 276% for total hip and 208% for total knee replacements by 2030, with current (direct and indirect) costs of over $5 billion/year expected to also rise [3,4]. With a dissatisfaction rate of between 20 and 30% after total hip and knee replacement (TJR) surgery [5], a rate of inappropriate surgery of 20–25% [6] and the lack of uniform capture of validated and relevant outcome measures, the question is not whether these procedures are cost-effective, which they are [7], but whether how and/or when TJR is utilized may be devaluing its true impact.

Decision aids are known to improve the efficiency of surgical decisions [8,9], but are more commonly constructed to predict immediate-term outcomes such as length of stay, need for rehabilitation and post-surgery complications [10]. This is likely due to the availability of these types of data. We have previously explored surgeons’ perceptions of utilizing decision aids to identify patients at risk of poor pain and functional outcomes (non-responders) following total knee replacement (TKR) [11]. We found that surgeons prioritise clinical acumen but do believe that decision aids can improve communication and patient-informed consent. Consequently, with data from our institutional registry, we constructed a prediction tool to help surgeons identify potential non-responders to total knee replacement (TKR) so that those who are deemed responders may be more efficiently processed for surgery [12], while those who are not would either be referred for non-surgical care or have their modifiable factors (obesity, mental health, co-and morbidities) optimised and further consideration given for surgery afterwards.

It is well established that total knee and total hip replacement recipients differ in their trajectory of recovery after surgery [13], with a tendency for people undergoing THR to reach their peak improvement in pain and function earlier than those undergoing TKR. However, maximum improvement is achieved for both within 1 year, albeit a higher proportion of THR recipients report greater pain and function improvements than those who undergo TKR [14]. The degree to which baseline patient characteristics predict pain and function outcomes varies across the literature [13,14,15], although factors such as baseline symptom and radiographic OA severity, obesity, mental health, and co-morbidities (variously measured) are consistently reported.

Given the variation in patterns of recovery following hip and knee TJR, we aimed to determine whether pre-operative prognostic variables that predict non-response to THR aligned with those that informed our previously constructed TKR nomogram [12] and therefore determine whether use of the existing tool could be interchangeable across procedures. Our end goal was to produce an internally validated nomogram that accurately predicted the likelihood of non-response to THR surgery at 1 year. We envisaged that whether existing or in the form of a new tool, a prognostic nomogram would be useful in informing targeted interventions for improving outcomes in people at risk of non-response to THR.

## 2. Patients and Methods

### 2.1. Study Setting and Participants

This study was undertaken at St. Vincent’s Hospital (SVH), in Melbourne Australia. All primary elective THR performed at SVH between January 2012 and December 2019, were assessed for study inclusion. A consecutive cohort of unilateral, elective, primary THR with baseline and 12 month follow-up data was included.

### 2.2. Data Collection

Data of individuals who had undergone THR between 2012 and 2019 were extracted from our institutional registry. The St. Vincent’s Melbourne Arthroplasty and Outcomes (SMART) registry is a clinical registry of all elective hip and knee joint replacements undertaken at the hospital from 1998 and was the same data source for our previously constructed TKR nomogram [12]. The SMART registry and data dictionary [16] have been previously described in detail. The registry holds an extensive array of prospectively collected demographic, surgical and clinical data, as well as patient-reported outcome measures. An osteoarthritis questionnaire (Western Ontario and McMaster Universities Arthritis Index (WOMAC)) and a general health questionnaire Veteran’s Rand Survey (VR12) are routinely collected within 12 weeks of surgery and at regular time points post-surgery, including at 1 year. Data are loaded onto the SMART registry by dedicated registry staff. Data quality audits are conducted annually, with missing and implausible data investigated by the research team and rectified where possible.

### 2.3. Surgery

All patients underwent an elective primary THR, performed or supervised by one of 19 surgeons. The use of either cemented or uncemented implants was determined by surgeon preferences and implants were used consistently throughout the study period.

### 2.4. Outcome of Interest

Our primary outcome was the likelihood of non-response to THR 1 year after surgery. We used the same process for deriving the primary outcome as we did when constructing our prior knee nomogram [12]. Changes in pain, function and global scores post-THR, relative to baseline, were derived from the Western Ontario and McMaster Universities Osteoarthritis Index (WOMAC), a self-administered measure of outcome of OA interventions, validated for use in THR [17]. We then used the OMERACT-OARSI responder criteria [18] to classify patients as either a responder or non-responder to THR based on the change in WOMAC scores. To ascertain whether a patient has achieved a meaningful response to THR, a normalised score (0 to 100) for each subscale of pain, function and global is created. Following OMERACT-OARSI criteria, a patient is considered a responder to THR if they report an improvement from baseline scores in either pain or function subscales of at least 50% and 20 points. If this threshold is not reached, then improvement in two of the three following criteria is also considered clinically meaningful; a minimum of 20% and 10 points in pain score; a minimum of 20% and 10 points in function score; and a minimum of 20% and 10 points in the global score. Patients who do not meet these thresholds are deemed non-responders and, according to OMERACT-OARSI criteria, have not achieved a clinically meaningful improvement from THR surgery.

### 2.5. Candidate Predictors 

Pre-operative prognostic variables were chosen a priori as candidate predictors based on clinical reasoning and prior research. Aside from age, sex and aetiology, candidate predictors were limited to those that were considered potentially modifiable. Potential covariates therefore included smoking status and body mass index, which was assessed as both a continuous variable and according to World Health Organisation criteria as non-obese (BMI < 30 kg/m^2^), obese class I (30 ≤ BMI < 35 kg/m^2^), obese class II (35 ≤ BMI < 40) and obese class III (BMI ≥ 40 kg/m^2^). The American Anaesthesiologist’ (ASA) Physical Status Classification (1–4) [19] and the Charlson Co-morbidity Index (CCI) [20] were used as co-morbidity measures, with the latter assessed as a continuous variable and by each individual Charlson co-morbidity. The pre-operative anteroposterior radiographs taken within three months of surgery were used to assess radiographic osteoarthritis severity and assign a Kellgren–Lawrence (K-L) grade (0–4) [21]. Quality of life was derived from the routinely collected VR12 baseline physical (PCS) and mental (MCS) component scores [22]. 

Patient postcodes were used to assign a Socio-Economic Index for Areas (SEIFA) score (1–10) [23] and a geographic accessibility index (ARIA+) [24], which serve as a proxy for socio-economic status and remoteness. The use of interpreter services as a measure of English language proficiency was also recorded, but these were not included as candidate predictors. As the purpose of this study was to develop a predictive nomogram for use as a decision support tool prior to undergoing surgery, procedural characteristics such as prosthesis type, fixation and surgical approach as well as post-operative length of stay (days), discharge disposition (home vs. rehabilitation) and adverse events (according to Clavien–Dindo criteria) [25] are presented for descriptive purposes only. 

### 2.6. Statistical Analyses

Categorical variables were presented as the frequency and percentage. Continuous variables were presented as the mean and standard deviation (SD) or the median and inter-quartile range (IQR). Aligning with the development of our knee nomogram [12], the statistical approach was a follows:Unadjusted and then adjusted logistic regression models clustered on the individual patient were run, with the candidate predictors described above analysed for associations with non-response to THR surgery at 1 year.Quadratic transformations were incorporated into the models to test for the linearity of association between candidate explanatory variables and the non-response endpoints.Multivariable models were then assessed for collinearity and potential interactions between pairs of candidate predictors were further tested.Overall fit of the model was tested using a Hosmer and Lemeshow goodness-of-fit test.The Akaike and Bayesian Information criteria were used to assess relative goodness of fit between multiple, competing multivariable model solutions prior to the selection of the final model for the development of the final prognostic nomogram.

#### 2.6.1. Nomogram Construction

Following the same method as previously described by Katten et al. [26], independent prognostic correlates of post-operative non-response that we observed in our adjusted modelling were used to derive a prognostic nomogram for non-response to THR.

#### 2.6.2. Nomogram Validation

As per our knee nomogram [12], internal validation was derived via the concordance index and evaluation of nomogram calibration. The concordance index captures the likelihood that a study case drawn at random from the analysis dataset that was classified as a non-responder before another case drawn at random records a higher likelihood of non-response on the prognostic nomogram. The index was ascertained from the original 2177 THR used to derive the multivariable model described above, with 500 bootstrapped random samples. A separate round of 500 bootstrapped resamples was then used for calibration of the nomogram. Cases were categorized in accordance with their non-response probabilities as predicted by the nomogram and the mean scores of these probability groups were compared to the empirically observed non-response estimates on a calibration curve. The calibration curve is representative of the degree of agreement between predicted and observed values across a spectrum of predicted probabilities of non-response.

All analyses were performed using R version 3.6.3 (R Foundation for Statistical Computing, Vienna, Austria).

## 3. Results

### 3.1. Study Population

A total of 2248 primary elective THR were performed in 1950 patients during the study timeframe, of whom all had complete baseline data. Eight cases (16 hips) of simultaneous bilateral THR and 20 cases who underwent revision surgery within 1 year were excluded. Thirty-five cases were missing a 12 month WOMAC questionnaire and therefore also excluded due to death (*n* = 25), declining to complete the survey (*n* = 6) or having relocated overseas (*n* = 4). This left 2177 of 2212 (98.4%) eligible cases with complete data for analysis. In total, 194,2177 (8.9%) cases failed to reach the threshold for achieving a clinically meaningful response to surgery as per OMERACT-OARSI definitions and were considered non-responders to THR.

Demographic and clinical characteristics of the cohort are outlined in Table 1. The mean (SD) age was 66.5 (11.7) years and 1218 (58.9%) were female. Just over eight percent of the cohort required the use of an interpreter and 41.9% resided in a geographically disadvantaged area (SEIFA score ≤ 5) according to postcode. The probability of non-response to THR was higher amongst those who required an interpreter or were from a relatively disadvantaged postcode (Table 1). 

The average length of stay was 4.5 (2.6) days and 1723 (79.1%) of cases were discharged home, whereas 454 (20.9%) cases required inpatient rehabilitation. Most adverse events were minor; either Clavien–Dindo grade 1 (9.0%) or grade 2 (7.3%). In those who experienced a grade 3 complication (*n* = 90; 4.1%), a higher proportion (*n* = 16) were non-responders; however, these numbers were relatively low. There was no difference in responder rates between those who did or did not incur an unplanned readmission (Appendix A).

### 3.2. Predictors of Non-Response

On unadjusted modelling (Table 2), obese class III (BMI ≥ 40 kg/m^2^) was associated with 1.87-fold the odds of non-response (uOR 1.87; 95% CI 1.18, 2.98), relative to non-obese (BMI < 30 kg/m^2^). ASA Class ≥ 3 was associated with 1.45-fold the odds of non-response (uOR 1.45; 95% CI 1.08, 1.96) and CCI scores of 1 (uOR 1.53; 95% CI 1.06, 2.20) and ≥2 (uOR2.36; 95% CI 1.64, 3.40) were associated with an increased odds of non-response to THR. To assess whether the association between CCI ≥ 2 and non-response to THR was due to multiple Charlson co-morbidities or severity of the Charlson co-morbidity, each individual Charlson comorbidity as defined by Charlson et al., 1987 [27] was assessed for probability of non-response (Appendix A). This revealed that cerebrovascular disease (uOR 2.29; 95% CI 1.29, 4.07), chronic pulmonary disease (uOR 1.72; 95% CI 1.05, 2.82), connective tissue disease (uOR 1.90; 95% CI 1.13, 3.21), diabetes (uOR 1.83; 95% CI 1.26, 2.66) and mild liver disease (uOR 2.44; 95% CI 1.06, 5.61) correlated with non-response to THR (Table 2). Kellgren–Lawrence (K-L) grade < 4 also correlated with increased odds of non-response to THR relative to K-L 4 (uOR 1.99; 95% CI 1.47, 2.69) and each 10-point increase in baseline WOMAC Global score correlated with a reduction in post-operative non-response (uOR 0.87; 95% CI 0.80, 0.94).

Two multivariable models were run (Appendix A), model one using the CCI score and model two using individual Charlson comorbidities, with the latter demonstrating a stronger association with probability of non-response to THR and therefore selected as the model of choice to construct our nomogram (Figure 1). The final candidate variables associated with an increased odds of non-response and selected for our nomogram (Table 2) were therefore obese class III (adjusted OR 1.88; 95% CI 1.16, 3.05), cerebrovascular disease (aOR 2.39; 95% CI 1.33, 4.30), chronic pulmonary disease (aOR 1.64; 1.00, 2.71), connective tissue disease (aOR 1.99; 95% CI 1.17, 3.39), diabetes (aOR 1.86; 95% CI 1.26, 2.75), liver disease (aOR 2.28: 0.99, 5.27) and K-L grade < 4 (aOR 1.89; 95% CI 1.39, 2.56). In contrast, each 10-point increase in pre-operative WOMAC Global score was associated with a 14% reduction in the odds of non-response (aOR 0.86; 95% CI 0.79, 0.94), when other predictors were controlled for. A worked example is provided as a Appendix A.

To estimate an individual’s likelihood of non-response to THR, match each response or level of predictor for each explanatory variable to the corresponding points (top line) and then sum to produce a total score. Match the total points to the corresponding probability of non-response scale (bottom line) to derive the final estimate. 

### 3.3. Prognostic Nomogram

Independent correlates of non-response taken from adjusted logistic model two described above, were used to derive a prognostic nomogram that estimated the likelihood of non-response based on this suite of explanatory variables (Figure 1). The degree of contribution of each explanatory factor to non-response nomogram points in descending order was cerebrovascular disease, connective tissue disease, K-L grade, BMI ≥ 40 kg/m^2^, diabetes, and baseline WOMAC Global. The model’s concordance index was 0.70. The calibration curve (Figure 2) is an illustration of how accurately the nomogram predicted a probability of non-response when compared with the actual dataset-observed outcomes.

## 4. Discussion

In this large cohort study of 2177 consecutive cases of THR, we identified a non-responder rate of 9%, which is lower than for our TKR cohort (15%) and this is in keeping with the literature, whereby improvements in pain and function and satisfaction with THR are generally higher than for TKR [28,29]. Whilst our non-responder rate for THR was comparatively lower than for TKR, at a cost of $26 k per procedure [30], selection of people who do not respond to TJR could be costing Australia up to $119 million per year in direct hospital cost alone, on what can be regarded as low-value and wasteful care. 

Both the THR cohort in the current study and TKR cohort [12] in our prior study shared some prognostic factors for non-response to surgery, including morbid obesity, and milder radiographic and clinical symptom severity; however, there were also some notable differences. Unique to our THR cohort, a patient’s comorbidity profile appeared to play an important role in predicting risk of non-response to surgery. We also found no association between pre-operative mental well-being and response to THR, which in contrast had featured strongly as a predictor of poor pain and function outcome in our TKR nomogram cohort [12]. These differences have important implications for tailoring non-operative therapy and pre-surgery optimisation programmes for people with advanced hip and knee OA who may be considering TJR.

Current Royal Australian College of General Practitioner (RACGP) guidelines [31] indicate that people who receive the best outcomes following TJR include those with well-controlled comorbidities, a BMI between 20 and 30, “good mental health status” and radiographic osteoarthritis severity of K-L grade 3 or 4. Yet our current study and the broader literature are not entirely congruent with these recommendations. For example, both our hip and knee nomograms suggest that a K-L grade < 4 is prognostic for poor response to surgery, and that BMI is only prognostic for poor pain and function outcomes in those who are morbidly obese [6,7,8,9,10,11,12]. 

Twenty eight percent of our THR cohort had a K-L grade 3 OA at the time of surgery, and eight percent were morbidly obese, indicating possible missed opportunities for guideline-endorsed [32] non-surgical treatment in a substantial proportion of this cohort. Although it is unclear from the current study what, if any, non-operative interventions were attempted by patients prior to presenting for surgery, it has been reported that 57% of Australians who present for TJR have not received appropriate first-line treatments such as exercise therapy and weight loss [33]. A substantial body of evidence does suggest that implementation of standardized evidence-based non-surgical therapies, even in those with established OA, can result in meaningful improvement in both pain and quality of life for many, as well as cost savings based on deferral or avoidance of TJR [34]. 

Morbid obesity poses a challenge to treat patients with advanced OA, with diet and education proving most effective in people with lower levels of obesity [35]. A recent randomised controlled trial of a new pharmacologic option appears safe and effective in those with severe obesity, although not specifically trialled in the OA population [36]. Our randomised clinical trial of bariatric surgery in patients with advanced knee OA demonstrated that one-third of severely obese patients (BMI ≥ 35 kg/m^2^) declined to proceed with planned TKR up to 5 years post-original consent for surgery, due to symptom improvement [37]. This suggests a role for surgical weight loss interventions in definitively managing advanced lower-limb OA in patients who are morbidly obese [38]. It is also worth noting, however, that more than 40% of our THR cohort had a BMI within the range of 30 and 39 kg/m^2^ and while this group may benefit from diet and exercise, delay in referral to an orthopaedic surgeon in this group may not be warranted based on our findings.

Several Charlson comorbidities predicted poor response to THR in the current study. Prior literature has shown that comorbidities, whether assessed individually or by using validated indices [39,40], are associated with poor functional outcome after THR. In people with OA, comorbidities such as respiratory and cardiovascular disease have been linked with lower physical activity levels, compared to those without significant comorbidities [41] and current evidence demonstrates that activity levels do not improve after THR [42,43]. This suggests a role for increasing physical activity in patients with significant comorbidities prior to undergoing THR to maximise post-operative outcomes. However, there is inadequate knowledge and motivation for adhering to physical activity recommendations among people with OA and concomitant comorbidities who undergo joint replacement surgery, which poses a barrier to change [44]. 

Aside from our final candidate predictors, there were several patient demographic variables that were associated with non-response to THR, but these were not included in our final model. Our decision not to include socio-economic status and English language proficiency despite a clear association with non-response to THR was based on our a priori decision to only include candidate predictors that were deemed modifiable. Our use of SIEFA as a marker of socio-economic status has limited applicability for individual patient-level risk prediction as it is a relative index of socio-economic advantage/disadvantage according to geographic location. Nonetheless, patients who present from geographically disadvantaged locations and/or poor English language proficiency may require assessment for additional community supports both before and after surgery.

There are several strengths to this study including the large sample size, the near-complete follow-up of the cohort, as well as a broad range of candidate predictors available for assessment. The methods used for assessing the internal validity of our tool were robust and the internal validation itself suggested good nomogram performance. However, best practice requires external validation to confirm the generalisability of prediction tools [45], which poses a challenge, given the lack of available and comparable datasets for TJR [46]. In that regard, external validity of our prognostic nomogram for knee replacement has been independently assessed and only partially supported [47], albeit in a dataset with notable differences in sample characteristics. While the external validation study [47] substantiated the importance of baseline radiographic and clinical symptom severity in predicting non-response to TKR, the roles of BMI and mental well-being (also predictors in our knee nomogram study) [12] were not supported. However, there were very few morbidly obese patients (*n* = 17) in the external validation cohort and the timeline for capturing mental well-being prior to surgery was variable, highlighting the limited scope when using study cohorts that do not mimic the source dataset.

It is worthwhile noting that while the registry used in this study contains a comprehensive array of variables, it is by no means exhaustive. It is therefore entirely possible that risk factors for poor response to THR exist that were not captured in the registry and therefore were not included in our model, although a recent systematic review of pre-operative predictors of THR outcomes does align with our findings [48]. We also excluded non-modifiable factors such as socio-economic status and the use of an interpreter from our model despite a higher rate of non-response to THR among these groups. We felt these factors are better addressed as part of surgery and discharge planning, rather than influencing the decision to refer to or proceed with surgery.

## 5. Conclusions

In this study, we observed several pre-operative patient characteristics that were associated with poor patient outcomes which informed the development of a prognostic calculator for estimating the likelihood of non-response to THR. We believe this is the first tool to predict non-response to THR based on patient-reported outcomes. Notability, we found that both the probability of and predictive prognostic factors for non-response to surgery differ between total hip and total knee replacement (TKR), confirming the benefit of designing decision support tools that are both condition and surgery site specific. Like our knee nomogram, with external validation, we envisage our nomogram for THR may serve as a useful tool that supports the shared decision-making process between patients and their allied health or general practitioners, when considering specialist referral or for use by surgeons to highlight with their patients opportunities for pre-surgery optimisation.

## Figures and Tables

**Figure 1 jcm-11-01649-f001:**
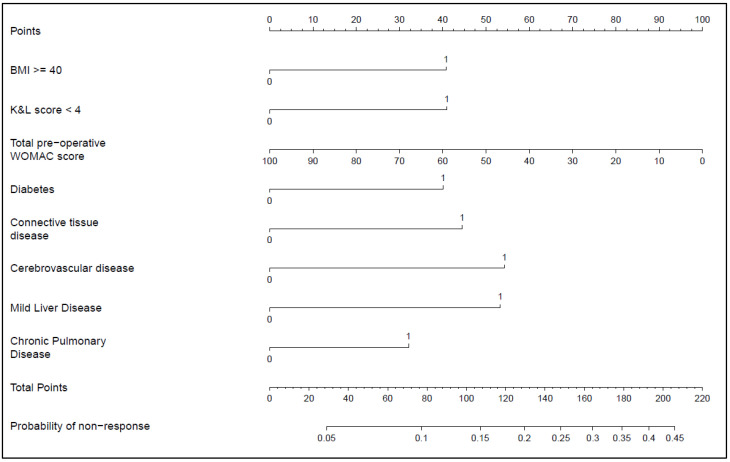
Nomogram for non-response to THR.

**Figure 2 jcm-11-01649-f002:**
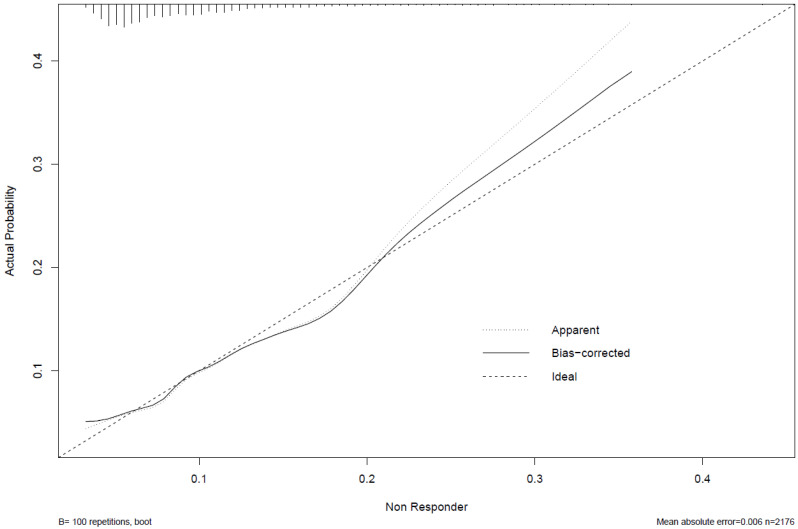
Calibration curve. The x-axis represents the nomogram-predicted probability, the y-axis represents the empirically observed probability and the dashed diagonal line depicts perfect agreement between the predicted and observed probability of non-response to THR.

**Table 1 jcm-11-01649-t001:** Baseline patient demographics and characteristics.

Variable	Overall (N = 2177)	Responder (N = 1983)	Non-Responder (N = 194)	*p*-Value
Sex [N (%)]				0.825
Female	1218 (55.9)	1108 (91.0)	110 (9.0)	
Male	959 (44.1)	875 (91.2)	84 (8.8)	
Age [mean (SD)]	66.5 (11.7)	66.3 (11.7)	68.0 (11.6)	0.051
^$^ BMI [mean (SD)]	30.5 (6.2)	30.4 (6.1)	31.3 (6.5)	
Obesity Class [N (%)]				<0.043
BMI < 30 kg/m^2^	1116 (51.3)	1026 (91.9)	90 (8.1)	
30 ≤ BMI < 35 kg/m^2^	612 (28.1)	560 (91.5)	52 (8.5)	
35 ≤ BMI < 40 kg/m^2^	286 (13.1)	258 (90.2)	28 (9.8)	
BMI ≥ 40 kg/m^2^	163 (7.5)	139 (72.9)	24 (14.7)	
^&^ ASA Score [N (%)]				0.013
≤2	1283 (58.9)	1185 (92.4)	98 (7.6)	
≥3	894 (41.1)	798 (89.3)	96 (10.7)	
Charlson Comorbidity [N (%)]				
0	1273 (58.8)	1187 (93.2)	86 (6.8)	<0.001
1	521 (23.9)	469 (90.0)	12 (10.0)	
≥2	383 (17.6)	327 (85.4)	56 (14.6)	
Smoker [N (%)]				0.307
Current	309 (14.2)	275 (89.0)	34 (11.0)	
Ex	616 (28.3)	567 (92.0)	49 (8.0)	
Never	1252 (57.5)	1141 (91.1)	111 (8.9)	
^+^ K-L Grade [N (%)]				<0.001
≤3	616 (28.3)	534 (86.7)	82 (13.3)	
4	1561 (71.7)	1449 (92.8)	112 (7.2)	
Aetiology [N (%)]				0.328
Osteoarthritis	1869 (85.9)	1711 (91.5)	158 (8.5)	
Inflammatory Arthritis	91 (4.2)	80 (87.9)	11 (12.1)	
Avascular Necrosis	146 (6.7)	129 (88.4)	17 (11.6)	
Dysplasia	71 (3.3)	63 (88.7)	8 (11.3)	
Pre ^ WOMAC-Pain [mean (SD)]	65.6 (18.2)	66.0 (18.0)	61.6 (19.7)	0.001
Pre WOMAC-Motion [mean (SD)]	70.0 (20.4)	70.3 (20.3)	66.9 (20.8)	0.028
Pre WOMAC-Function [mean (SD)]	68.2 (17.5)	68.6 (17.4)	64.6 (17.7)	0.002
Pre WOMAC-Global [mean (SD)]	67.8 (16.7)	68.2 (17.2)	64.1 (17.2)	0.001
Pre-surgery VR12 * MCS [mean (SD)]	41.1 (14.9)	41.2 (15.0)	39.9 (14.1)	0.226
Pre-surgery VR12 ^#^ PCS [mean (SD)]	24.3 (7.4)	24.4 (7.4)	23.9 (6.9)	0.387
Interpreter Required				0.025
Yes	178 (8.2)	170 (87.6)	24 (12.4)	
No	1999 (91.8)	1829 (91.6)	154 (8.4)	
Socio-Economic Index				0.011
≤5	913 (41.9)	815 (89.3)	98 (10.7)	
≥6	1264 (58.1)	1168 (92.8)	96 (7.2)	
Rurality [N (%)]				0.428
Metropolitan	1731 (79.5)	1581 (91.3)	150 (8.7)	
Regional	446 (20.5)	402 (90.1)	44 (9.9)	

^$^ BMI = body mass index; ^&^ ASA = American Society of Anaesthesiologists; ^+^ K-L = Kellgren–Lawrence; ^ WOMAC = Western Ontario and McMaster Universities Arthritis Index; * MCS = Mental Component Scale; ^#^ PCS = Physical Component Scale.

**Table 2 jcm-11-01649-t002:** Predictors of non-response to THR.

Factor	Level	uOR (95% CI)	*p*-Value	aOR (95% CI)	*p*-Value *
Obesity Class III	<40 kg/m^2^	Reference		Reference	
	>40 kg/m^2^	1.87 (1.18, 2.98)	0.008	1.89 (1.16, 3.07)	0.010
^ Diabetes		1.83 (1.26, 2.66)	0.002	1.86 (1.26, 2.75)	0.002
^ Chronic pulmonary disease		1.72 (1.05, 2.82)	0.030	NS	
^ Connective tissue disease		1.90 (1.13, 3.21)	0.016	1.95 (1.14, 3.33)	0.014
^ Cerebrovascular disease		2.29 (1.29, 4.07)	0.005	2.39 (1.33, 4.30)	0.004
^ Mild liver disease		2.44 (1.06, 5.91)	0.036	NS	
K-L grade	4	Reference		Reference	
	≤3	1.99 (1.47, 2.69)	<0.001	1.91 (1.41, 2.61)	<0.001
Pre-op WOMAC Global ^		0.87 (0.80, 0.94)	0.001	0.86 (0.79, 0.94)	<0.001

* Hosmer and Lemeshow: *p* > 0.05; ^ per 10 units; ^ comorbidities derived from the Charlson Comorbidity Index.

## Data Availability

Consent for data sharing was not obtained and ethics approval would be required from the study institutions for future use of these data.

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
