# Peer review of "A Nomogram for Predicting Non-Response to Surgery One Year after Elective Total Hip Replacement"

_jcm, 2022, doi:10.3390/jcm11061649_

Round 1
Reviewer 1 Report
Interesting work conducted with good methodology, however, he recommends changes1) In the abstract the endpoints do not match the text
2) The quick description of WOMAC and OMERACT-OARSI is missing in the data collection
3) The quick description of WOMAC and OMERACT-OARSI is missing in the data collection
4) Tab 1: what is meant by PCS and PCS?
5) Check tab 2
6) Specifies in tab 2 that the comorbidities are from the Charlson Index
7) In the discussion in line 290- 295 it is appropriate to add the effectiveness of hip infiltrative therapy with the possibility of using different hyaluronic acids, PRP, stem cells (Hybrid Hyaluronic Acid versus High Molecular Weight Hyaluronic Acid for the Treatment of Hip Osteoarthritis in Overweight/Obese Patients Scaturro, D., Vitagliani, F., Terrana, P., ...Midiri, M., Mauro, G.L.; , Intra-articular hyaluronic acid injections for hip osteoarthritis Mauro, G.L., Scaturro, D., Sanfilippo, A., Benedetti, M.G., The effectiveness of intra-articular injections of Hyalubrix® combined combined with exercise therapy in the treatment of hip osteoarthritis Mauro, G.L., Sanfilippo, A., Scaturro, D.)
8) I find the conclusion too absolute. The nomogram should be supplemented with other information
Reviewer 2 Report
Thank you very much for reporting the results of your wonderful research.
This is a very nice paper. Therefore, we believe that the data and structure is fine as it is. I understand the content, but I did not understand how this tool can be used by those who read this paper. It does not describe how to apply the nomogram proposed in this paper to a patient in front of the reader and determine whether the patient should undergo TKR or not.
Is it safe to assume except obesity above BMI 40, diabetes, chronic lung disease, connective tissue disease, liver disease, KL grade, and preoperative WOMAC did not affect THR outcomes?
If authors could add a new chapter that elaborates on this point, I think it would be of good quality.
Round 2
Reviewer 1 Report
Dear authors, the changes have been made correctly.The paper is clear and the topic interesting
Reviewer 2 Report
Thank you very much for taking the time to make this correction. I believe that the paper should definitely be published as is. I look forward to your continued guidance in the future.
This manuscript is a resubmission of an earlier submission. The following is a list of the peer review reports and author responses from that submission.